# A Case Study of a Tiny Machine Learning Application for Battery State-of-Charge Estimation

Spyridon Giazitzis [1], Maciej Sakwa [1,*], Sonia Leva [1], Emanuele Ogliari [1], Susheel Badha [2] and Filippo Rosetti [2]

[1] Department of Energy, Politecnico di Milano, 20156 Milan, Italy; spyridon.giazitzis@polimi.it (S.G.); sonia.leva@polimi.it (S.L.); emanuelegiovanni.ogliari@polimi.it (E.O.)

[2] Infineon Technologies, 9500 Villach, Austria; susheel.badha@infineon.com (S.B.); filippo.rosetti@infineon.com (F.R.)

\* Correspondence: maciej.sakwa@polimi.it

**Abstract:** Growing battery use in energy storage and automotive industries demands advanced Battery Management Systems (BMSs) to estimate key parameters like the State of Charge (SoC) which are not directly measurable using standard sensors. Consequently, various model-based and data-driven approaches have been developed for their estimation. Among these, the latter are often favored due to their high accuracy, low energy consumption, and ease of implementation on the cloud or Internet of Things (IoT) devices. This research focuses on creating small, efficient data-driven SoC estimation models for integration into IoT devices, specifically the Infineon Cypress CY8CPROTO-062S3-4343W. The development process involved training a compact Convolutional Neural Network (CNN) and an Artificial Neural Network (ANN) offline using a comprehensive dataset obtained from five different batteries. Before deployment on the target device, model quantization was performed using Infineon's ModusToolbox Machine Learning (MTB-ML) configurator 2.0 software. The tests show satisfactory results for both chosen models with a good accuracy achieved, especially in the early stages of the battery lifecycle. In terms of the computational burden, the ANN has a clear advantage over the more complex CNN model.

**Keywords:** state of charge; TinyML; battery; deep learning; IoT





## 1. Introduction

Progressing electrification in the transportation sector is nowadays incentivized by various policies to boost the penetration of Renewable Energy Sources (RESs) and minimize the use of fossil fuels [1]. An increased capacity of Lithium-Ion (Li-ion) batteries is the necessary solution to the mismatch between the power load and generation from RESs [2], which makes them an interesting topic to many researchers. However, apart from the development of batteries themselves, it is important to create a sophisticated and intelligent Battery Management System (BMS), as they play a critical role in enhancing energy efficiency and ensuring battery protection by dealing with tasks such as balancing the charge levels of cells, monitoring the temperature, and regulating the current and voltage.

BMSs should be able to estimate several important parameters, such as State of Charge (SoC), State of Health (SoH), and Remaining Useful Life (RUL), that cannot be measured directly through sensors [3]. In particular, precise estimation of the SoC is crucial for providing accurate information about the energy level of the battery, preventing over/under charging, power failures, voltage imbalances, and thermal runaway, which can result in fire or explosion in batteries [4–7]. Thus, many thermal runaway prevention and mitigation strategies [8] have been developed considering the battery's SoC. Moreover, precise SoC control can enhance the battery life and generally improve the overall battery performance, as it mitigates additional stress on the batteries that leads to chemical reactions degrading the battery materials [9].

The SoC is defined as a ratio of the available capacity of the battery to the maximum possible capacity, given a specific degradation state [10]:

$$SoC = \frac{Q_{cur}(t)}{Q_{max}(cycle)} \cdot 100\% \tag{1}$$

where $Q(t)_{cur}$ is the actual current battery capacity at a given point in the discharge cycle and the $Q_{max}$ is the maximum battery capacity in the studied discharge cycle. In contrast, the SoH is defined as a percentual ratio of the maximum available capacity of the battery in a specific degradation state to the maximum nominal capacity [11]:

$$SoH = \frac{Q_{max}(cycle)}{Q_{max}(nominal)} \cdot 100\% \tag{2}$$

where $Q_{max}(cycle)$ is the maximum available capacity of the battery in the discharge cycle and the $Q_{max}(nominal)$ is the maximum nominal capacity at the beginning of the lifecycle. In relation to the SoH, the RUL can be defined as the remaining number of charge/discharge cycles until the battery falls under the SoH threshold that warrants a replacement. Taking electric vehicles (EVs) as an example, this threshold is usually defined at 80% of the SoH, under which battery degradation might cause unpredictable behavior and faster-than-expected discharging. In the field of BMSs designed for EVs, it is vital to accurately estimate all three parameters. In this way, a complete picture of the current degradation state of the battery can be obtained. Furthermore, the corresponding maximum driving range and the actual current driving range can be easily found and communicated to the user.

### 1.1. State-of-the-Art SoC Estimation

As described, the SoC has to be estimated based on other measurements, such as voltage (V), current (I), and temperature (T) [12]. In Figure 1, an overview of SoC estimation methods is presented, which are normally applied to accurately estimate the SoC. It should be emphasized that laboratory methods for estimating the SoC can only be effectively utilized under specific conditions and with knowledge of certain initial battery parameters to achieve satisfactory results [13]. However, they are not feasible in real-time applications due to measurement offsets and a potentially noisy environment that could compromise measurement accuracy.

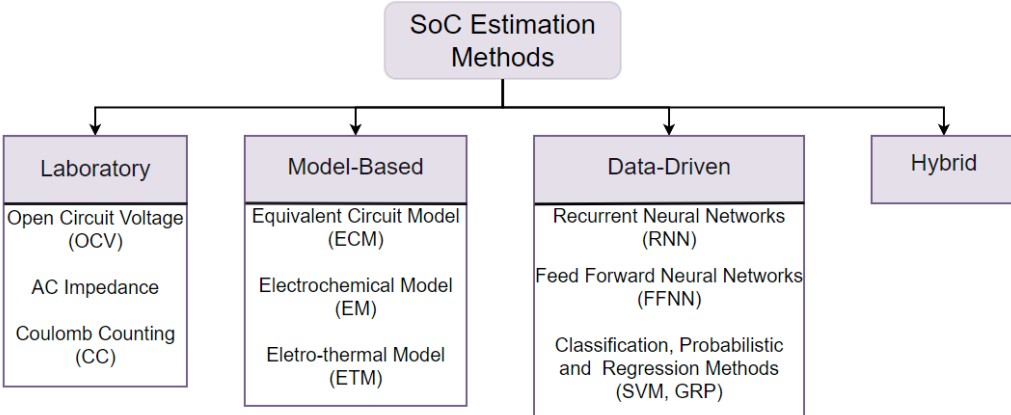

**Figure 1.** Different approaches to SoC estimation.

Various physical-model-based approaches have been used, such as electrochemical models [14], equivalent circuit models [15], and electro-thermal models [16]. To achieve a satisfactory SoC estimation accuracy, these models rely on partial differential equations and/or complex electrical circuits for describing electrochemical reactions/phenomena,

such as porous electrode theory, lithium diffusion, and polarization [17]. Furthermore, the parameters of the elements in circuit models can change due to several external factors such as the battery temperature and degradation level, so an additional algorithm would have to be developed to update these parameters. For this reason, to improve the accuracy and robustness of these models, algorithms based on filters (Kalman, particle, H infinity) and observers (Luenberger) are also included [18]. In [19], the authors developed a model-based method for SoC estimation based on an improved adaptive particle swarm filter to enhance its accuracy and robustness, achieving an error of less than 1%. Moreover, they mitigate the error generated due to the initial SoC offset and battery aging. On the other hand, in [20], an improved physics-based equivalent circuit model combined with an extended Kalman filter was developed for estimating the SoC. The pseudo-two-dimensional model was modified to improve the relationship between the electrochemical and the electrical parts and achieved an SoC estimation error of less than 1%. Overall, model-based methods achieve low errors when the parameters of the model and initial conditions are accurately calculated. This leads to a time-consuming procedure that requires excessive memory and computational requirements, making them inefficient for real-time applications [21].

On the other hand, due to the advancement in computational power of the devices and increased availability of historical data, data-driven methods are more often chosen as a suitable solution for SoC estimation. The developed models can be used as a 'black-box' to perform SoC estimation with a low error and without the need to solve any PDE or create a complicated ECM. Numerous data-driven approaches have been tested up to date, with methodologies focusing both on Machine Learning (ML) techniques such as Gaussian Process Regression or Support Vector Machines, and Deep Learning (DL)-related Neural Networks (NNs) [22]. Thorough reviews of the available methods paired with detailed performance analyses can be found in [23,24]. Examples of applications include [25], where the authors developed a temporal convolutional network for SoC estimation, which is mainly based on stacked CNN layers and dilation, to control kernel spacing, capturing temporal dependencies and achieving an error of less than 1% and outperforming RNN models such as LSTM and GRUs. Moreover, in [26], a bidirectional long short-term memory neural network was developed, where its parameters were optimized through the Bayesian optimization algorithm. The model achieved SoC estimation with an accuracy of around 1% for both MAE and RMSE metrics.

While data-driven methods offer ease of implementation and high performance for BMS applications, they are not universally suitable. In some scenarios, their deployment might be impractical. A key challenge of data-driven approaches lies in their potential for parameter explosion, particularly with Convolutional Neural Networks (CNNs). This significant parameter count restricts deployment to powerful computers or cloud-based solutions, limiting their applicability in resource-constrained environments. For instance, a BMS deployed on an EV would likely operate remotely without consistent high-bandwidth internet access, rendering cloud-based solutions infeasible. However, data-driven methods have the advantage of being adaptable to direct sensor readings (e.g., battery voltage or current) and require a minimal cell-specific configuration.

To address the limitations of data-driven methods in remote settings while leveraging their direct data usage capabilities, a solution exists: Tiny Machine Learning (TinyML). TinyML techniques can be applied to reduce the model size and memory footprint, enabling deployment on resource-constrained embedded devices suitable for remote locations. This process typically involves a trade-off between model size and performance, with a slight decrease in accuracy in exchange for significant memory savings.

### 1.2. TinyML for SoX Etimation

TinyML has been identified as one of the most promising frontiers in data-driven SoC estimation, with embedded edge sensor devices used to create smart battery packs that can conduct a real-life evaluation of the performance and states of the battery [27]. The challenge of designing and optimizing an ML model on low-power Internet of Things

(IoT) devices comes with a lot of benefits, including energy efficiency, low cost, low latency, and the ability to perform local data processing, avoiding unnecessary data transfers [28]. Despite numerous advantages, there are some drawbacks as the devices often operate in uncontrolled environments with unpredictable surrounding conditions and energy supply [29]. An additional challenge arises regarding the ML model. SoC estimation is a complex and multivariate task as the value heavily depends on the battery type, temperature, and current aging state. Moreover, there are severe limitations to the model and algorithms that can be implemented on IoT devices due to constraints on memory and computation power available [30,31]. Therefore, the task includes a process of optimization as the desired outcome is a TinyML model that is small enough to fit on low-power IoT devices and robust enough to handle the intricacies of SoC prediction for various test cases.

Generally, ML and DL models used for offline SoC prediction are well defined and achieve satisfactory results with a mean error within a range of 2% to 3% [32–34]. However, the IoT implementations of the models are a relatively unexplored area and are gaining increasingly more popularity due to their high performance. However, some related studies that tackle the same problem from a different or similar perspective have been identified and will be briefly explored.

In a previous study [12], the authors identify the CNN and Gated Recurrent Units (GRUs) as the most promising models in terms of quantization. A comparative study is performed with two different post-training quantization methods. The authors achieve splendid results, with the MAE below 2% for the CNN model. However, the training and test procedure is based on a single charge and discharge cycle. In [35], the authors perform an estimation of the SoC based on real-life operation data from a BMWi3 electric vehicle (EV). Ten highly correlated features were selected after a sensitivity analysis with good results achieved for simple DL models. An inference time study was performed to estimate the necessary size of the computational unit post-quantization. In [36], the authors perform a similar study but for the SoH prediction using recorded battery parameters (V, I, and T), with the best performance achieved for a hybrid CNN-GRU model. The models were tested both pre- and post-quantization and no significant changes to performance were observed. A qualitative comparison between the identified related studies that utilize TinyML for SoC and SoH (SoX) predictions and the one presented in this paper can be seen in Table 1.

**Table 1.** Qualitative assessment of the identified related research papers that target SoX predictions on TinyML devices.

| Authors | Model | Variable | Quantization | Dataset | Result |
|---|---|---|---|---|---|
| Mazzi et al. [12] | CNN, GRU | SoC | Post-training (TFLM, STM32.AI) | Private, small (350 k samples) [I, V, T] data | CNN outperforms GRU with an RMSE of 2.36% for ref. models with a drop to 4.97% for quantized models. |
| Pau et al. [35] | ANN, CNN, LSTM, GRU | SoC | Post-training (SPC5-STUDIO.AI) | Private, Real EV use data, multi-feature | ANN outperforms other models with an RMSE of 1.95% for the ref. model with 10 features. No data on post-quantization performance. |
| Crocioni et al. [36] | CNN, LSTM, CNN-LSTM, GRU, CNN-GRU | SoH | Post-training (TFLM, STM32.AI) | NASA [37], large (3.7 M samples), [I, V, t, T] data | CNN-GRU model outperforms others with an RMSE of 4.88% for the ref. model with a 0.5% drop for the quantized model. |
| Present study | ANN, CNN | SoC | Post-training (IFX) | Published, large (7.5 M samples), [I, V, T] data | ANN outperforms CNN with an RMSE of 3.79% for ref. models with a drop to 3.89% for quantized models. |

Considering the present study, the approach itself is most similar to the one presented in [12] with a three-feature time series for SoC estimation. However, it is the first time that such a large and complete dataset has been utilized for the purpose of TinyML-based SoX evaluation. Moreover, the proposed models achieve lower drops in accuracy from the quantization procedure, indicating that the proposed architecture is better suited for TinyML applications.

### 1.3. Paper Objective

In this paper, a new battery dataset will be used to train and test two NN architectures, namely a CNN and an ANN. The models will be trained offline (on a desktop) and subsequently quantized and optimized to be tested on the target IoT device. The conversion process will be carried out using Infineon MTB-ML 2.0 software and the quantized model will be implemented on the PSoC6 (CY8CPROTO-062S3-4343W) target device. The new dataset allows for a test spanning a variety of charge and discharge cycles and helps to better evaluate the potential of robust IoT implementations of SoC predictions.

This paper is structured as follows: first, the used datasets and case study are described in Section 2, and the used DL model and techniques are later described in Section 3, followed by a discussion o details of the model desired deployment board in Section 4. Preliminary results of the proposed model are presented in Section 5 and final conclusions in Section 6.

## 2. Dataset

SoC and SoH predictions are based on generally available open-source datasets such as the Center of Advanced Life Cycle Engineering (CALCE) [38] or NASA Li-ion Battery Aging Dataset [37]. In this case, a new online available dataset [39] has been used to run the training and test. The dataset is composed of recordings of six cylindrical, LG 2.5 Ah 18650 NMC batteries recorded in various temperature ranges. The discharging profiles follow the UDDS, US06, and LA92 driving cycles and six random combinations of these (cycles Mixed1-Mixed6). These driving profiles were chosen due to the high variety of driving patterns they can simulate. For example, UDDS simulates an urban route with an average speed of 31 km/h, while LA92 and US06 simulate more aggressive driving profiles, reaching an average speed of 39.6 km/h and 77.9 km/h. This is crucial, as in [40], the authors proved the importance of utilizing realistic diving profiles as they accelerate the degradation of the battery. This effect is not solely caused by the temperature, the depth of discharge, or calendar aging. Furthermore, for recharging the batteries, constant-current constant-voltage profiles were utilized. With a sampling rate of 10 Hz, the dataset is composed of over 7.5 million samples. The detailed parameters of the recorded data can be seen in Table 2. Overall, the new open-access battery dataset examines several battery cells and contains increased historical information for several features such as the cell current, voltage, and temperature. Moreover, due to different temperature scenarios and complex driving profiles, the new dataset provides more holistic information compared to previously studied and open-access datasets, enhancing the complexity and robustness of the trained models.

**Table 2.** Dataset main features.

| Features | Dataset |
| :---: | :---: |
| Battery Type | LG 2.5Ah 18650 NMC |
| Testing Equipment | Neware battery tester |
| Temperature Range | −20 °C, −10 °C, 0 °C, 10 °C, 25 °C, 35 °C |
| Driving Profiles | UDDS, US06, LA92, Mixed1-Mixed6 |
| Sampling rate | 10 Hz |
| No. of Samples | 7,676,032 |

To reduce the training time and the computation load of the models, the datasets are resampled to a 1 s mean. The data from the first five cells are used solely for the training of

the designed DL models, and the data from cell 6 are used for the test. As for the driving cycles, the recorded cycles vary from cell to cell, so the dataset is not fully consistent. For example, for cells 1–5 used in the training procedure, UDDS, US96, and Mixed1-Mixed4 profiles were utilized, while for cell 6, LA92 and Mixed5-Mixed6 were used.

For the case study, two driving profiles (1 and 2) were chosen out of the test dataset to compare the accuracy and the hardware requirements of the trained models based on the same driving cycle profile (LA92) but with different degradation states (around 100th and 400th cycles). In Figure 2, the SoC of the selected discharge profiles is depicted and the effect of the degradation is noticeable. Driving profile 1 (from here on called cycle 100) represents the main use scenario of the developed model before the second life of the battery should begin. In other words, for cycle 100, the SoH of the battery is far above the value of 80% when a replacement is usually warranted. In contrast, driving profile 2 (from here on called cycle 400) represents an edge case close to the necessary replacement of the battery, with SoH values close to 80% for most of the tested battery cells. However, the designed models are not informed on the stage of life of the battery (neither the SoH, the RUL nor the degradation state), as the designed approach is purely data-driven.

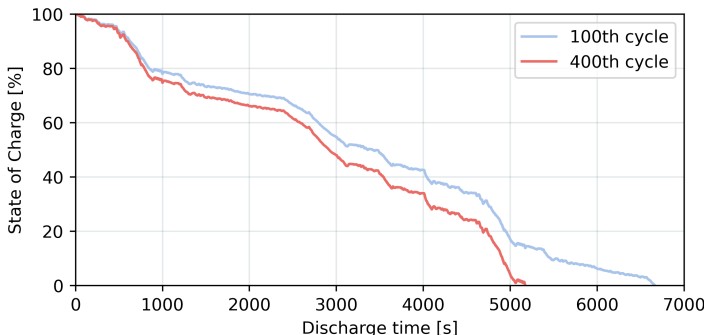

**Figure 2.** SoC values of testing profiles 1 and 2.

## 3. Model Definition

### 3.1. Tested Models

Two different models were trained and tested using the dataset described in Section 2 with hyperparameters optimized through a Bayesian optimization (BO) approach [32]. BO is an effective method for tuning model hyperparameters, known for its efficient performance in large hyperparameter spaces, where it performs faster compared to more exhaustive methods like random or grid search. BO uses probabilistic models to estimate the objective function and updates its assumptions to focus on promising areas of the hyperparameter search. The process is initiated by setting a prior distribution based on any existing knowledge of the parameters. As it progresses, BO intelligently tests new configurations, learning from the results of previous tests. This iterative process helps it to quickly converge on the best hyperparameters [41]. However, as the desired application relies on TinyML, the choice of available hyperparameters (number of units and filters) and activation functions is limited due to quantization and software restrictions.

The DL models were trained to use historical sensor-based data of the battery output [I, V, T] to predict the current SoC value. Generally, the prediction task can be described as:

$$SoC_n = f([I_{n-t}, V_{n-t}, T_{n-t}], \ldots, [I_n, V_n, T_n]) \tag{3}$$

where the SoC at a given second is a function of the historical data from a selected period. After a sensitivity analysis, the length of this period was set to 60 s as it offers the best performance in terms of accuracy with the additional advantage of convenience for real-time applications. Considering that three input features are studied at every point (I, V, and T), the input matrix has a shape of 3 by 60 samples. This matrix is subsequently flattened in

the first step of DL model processing. Figure 3 presents an additional visualization of the prediction procedure.

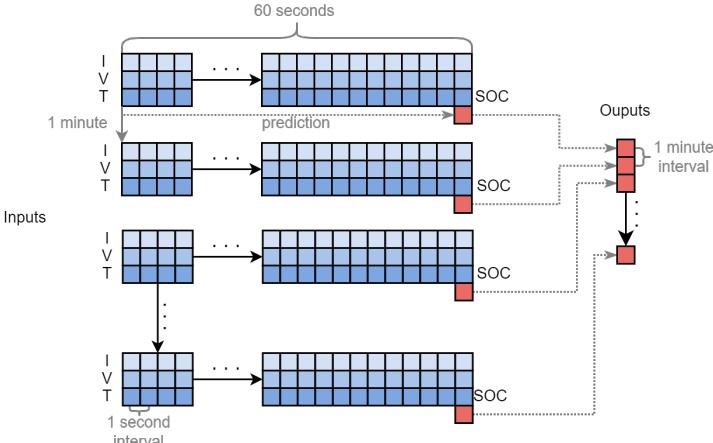

**Figure 3.** Prediction process—a tensor of 60 [I, V, T] values corresponding to 60 s of data is used as the input for the model (in blue) to predict the value of the SoC at the 60th s (in red). The output can be described as a time series of SoC data with a 1-min sample rate.

- Artificial Neural Network (ANN)—a standard fully connected feed-forward Artificial Neural Network was used as a universal benchmark model to be uploaded and tested on the device. The most basic building block of an ANN is called a perception (or a neuron) and an example can be seen in Figure 4.

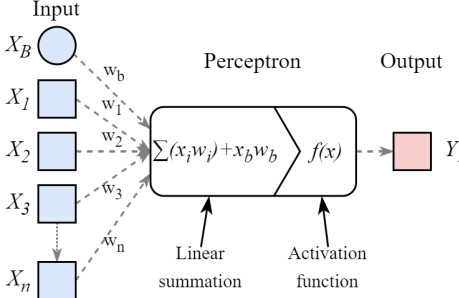

**Figure 4.** A single perception mathematical model.

An ANN is normally composed of multiple linear layers of neurons (including the input layer, one or more hidden layers, and an output layer) that sequentially pass the input feature tensor to the output, applying the selected transformations according to the equation:

$$y_i = f_i(\mathbf{W}_i^T x_i + b_i) \tag{4}$$

where $y_i$ is the output of the i-th layer, $f_i(z)$ is the activation function, and $W_i$ and $b_i$ are weights and bias matrices, respectively. This allows the ANN to model complex non-linear dependencies between the input and output. ANNs are generally used in a variety of tasks due to their inherent simplicity and flexibility. The main parameters that have to be set are the number and size (in terms of neurons) of hidden layers and the activation function that decides the output of neurons. The sizes of the input and output layers are usually imposed by the structure of the available data and the desired model output. The used network is composed of three linear layers with a decreasing number of neurons (34, 8, and 1, respectively) to obtain the SoC prediction at the output. All the layers use an ReLU activation function due to quantization limitations. In total, the selected structure results in 6.5 thousand trainable parameters.

- Convolutional Neural Network (CNN)—Convolutional layers are widely used for feature extraction purposes of complex inputs, such as images or audio recordings [42]. It is achieved through a procedure composed of *convolution* and *pooling* operations. In convolution, a kernel is passed over the input and multiplied (or convolved) with its corresponding segments to create a feature map (as seen in Figure 5). In reality, a number of kernels (also called filters) are passed over the input, each with a different set of weights, resulting in a number of extracted feature maps. The number of filters can be easily tuned depending on the complexity of the input. In the following step, the feature maps are reduced in size through the function specified in the pooling layer. While it is true that a pooling procedure can be considered destructive to the data due to severe downsampling, in the case of TinyML applications, it is all the more important to limit the memory footprint of the model. A single max-pooling layer halved the number of parameters and therefore occupied flash memory in the target device. Moreover, the inclusion of a pooling layer removes the need to flatten the input array as it is needed in the case of an ANN. The extracted features can be later fed into various typologies of models. In this case, it is a standard linear CNN with a single 1D convolution layer (with 16 filters) followed by a max pooling layer. The extracted feature maps are later flattened and passed through a series of three linear layers with a reducing number of neurons (52, 4, and 1 neuron, respectively) to obtain the SoC prediction at the output. All the layers use an ReLU activation function due to quantization limitations. In total, the selected structure results in 24.5 thousand trainable parameters.

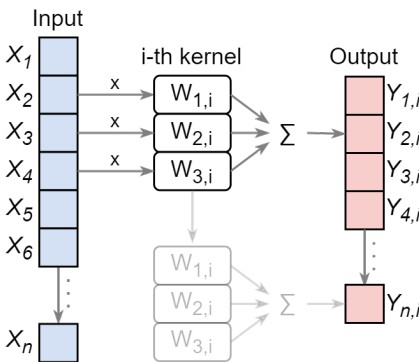

**Figure 5.** One-dimensional convolution operation.

*3.2. Evaluation Metrics*

Several evaluation metrics were used for model comparison. Additional metrics that describe the model performance in terms of computational and memory requirements are provided directly through Infineon MTB-ML 2.0 software. In detail, the mean absolute error (MAE) and the root mean square error (RMSE) were used to estimate the model performance.

$$MAE = \frac{1}{n} \sum_{j=1}^{n} |SoC_i - S\hat{o}C_i| \tag{5}$$

$$RMSE = \sqrt{\frac{1}{n} \sum_{i=1}^{n} (SoC_i - S\hat{o}C_i)^2} \tag{6}$$

where $SoC_i$ and $S\hat{o}C_i$ are the real value and the predicted values for each model, respectively.

## 4. Target Device

*4.1. IoT Device*

The target device that will be used for ML and DL model deployment is the CY8CPROTO-062S3-4343W PSoC 62S3 (Figure 6), which is a low-cost hardware platform

that enables the design and debugging of PSoC6 MCUs. The main parameters of this device are as follows:

- MCU—PSoC 6.
- Voltage range—1.8–3.3 [V].
- Flash memory—512 kB.
- SRAM—256 kB.
- WiFi + Bluetooth—Murata LBEE5KL1DX module (based on a CYW4343W combo device).

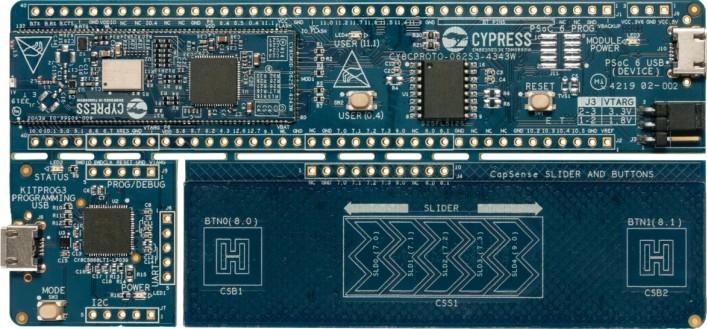

**Figure 6.** CY8CPROTO-062S3-4343W PSoC 62S3 target device.

To deploy the models onto the target device, MTB-ML software by Infineon was used. It provides a dedicated configurator for importing pre-trained DL models and generating embedded models for the target device. Furthermore, the optimized models are validated to provide information on the performance and resource requirements, such as the number of cycles to run inference, the memory requirements for the model weights (flash memory), or inference working memory (SRAM). In Table 3, a short description of the main supported features of MTB-ML can be seen.

**Table 3.** Features supported by the MTB-ML quantization engine.

| Features | Values |
|---|---|
| Formats | TFLite and H5 |
| Inference engines | TensorFlow Lite for microcontrollers Infineon inference engine |
| Core NN kernels | LSTM, GRU, MLP, Conv1d, Conv2d |
| Support NN kernels | flatten, dropout, reshape, input layer |
| Activations | relu, softmax, sigmoid, linear, tanh |
| Input data quantization level | 32-bit float, 16/8-bit integer |
| NN weight quantization level | 32-bit float, 16/8-bit integer |
| Cycle and memory estimation | Yes |

*4.2. Quantization*

Quantization reduces computational and memory costs during inference of ML/DL models by converting high-precision data (weights and inputs) to a lower precision (e.g., 8-bit integers). Quantization has several advantages, such as a reduction in the model memory footprint, that are vital for implementing models on embedded devices that possess low computation and memory capabilities (such as smartphones, smart sensors, and other IoT devices). Moreover, it improves the inference time due to a lowered computational complexity and enhances energy efficiency by lowering power consumption, making it ideal for real-time applications. However, quantization usually comes at the cost of a reduced prediction accuracy due to the reduced bit precision and noise introduced during the model (or layer) scaling. There are two main approaches to quantization: post-training quantization and quantization-aware training [43]:

- Post-training quantization (PTQ)—This compresses the weights or both weights and activations for faster interference. The process is performed without the need to retrain the model. It is a simpler and faster method that can be performed even with limited data, and it is widely used in existing embedded hardware solutions [44].
- Quantization aware training (QAT)—This introduces quantization during the model training or fine-tuning. It requires vast amounts of additional labeled data to accurately estimate the quantized weights. The fine-tuning results in some benefits, as the network can better adapt to the noise of quantization and achieve a better prediction accuracy compared to PTQ even for low-bit solutions [45].

In this paper, PTQ is applied using MTB-ML software due to its fastness and ease of use. Technically, PTQ is performed by mapping the floating point variables of the original model weights and activations to an integer grid with a size defined by the bit width *b*. This is achieved by defining a *scale factor s* and *zero-point z* of the quantized layer and modifying the weights from real variables (floating point) to integer representations calculated as [46]:

$$x_{int} = clip\left(\left\lfloor \frac{x}{s} \right\rceil + z; 0, 2^b - 1\right) \tag{7}$$

where $\lfloor \cdot \rceil$ is the rounding-to-nearest operation and clipping is the process of limiting a value to a range between the defined minimum and maximum values (which is 0 and $2^b - 1$ in this case). Therefore, the real value can be approximated from the integer value through *dequantization* as:

$$x \approx \hat{x} = s(x_{int} - z) \tag{8}$$

As a consequence, PTQ introduces noise to the model predictions due to the rounding and clipping operation errors induced through the quantization procedure. These errors strictly depend on the selected parameters and particularly on the quantization bit width *b*. The effects of a different *b* selection on the proposed models are later studied in Section 5.

MTB-ML supports two different quantization engines, the standard *TFLM* which is a part of the TensorFlow Deep Learning ecosystem, and *IFX* offered by Infineon MTB-ML 2.0 software. The details of the supported quantization precisions can be seen in Table 4. Moreover, during PTQ, the MTB-ML 2.0 software performs advanced scratch memory optimization that further reduces the inference time with minimal impact on the model precision. In this research paper, the IFX engine was selected to perform PTQ to better evaluate and compare the performance and accuracy loss of different quantization precisions.

**Table 4.** Quantization type supported by IFX quantization engine compared to the TFLM.

| Quantization Type | Input Data | Weights | Inference Engine |
|---|---|---|---|
| int8x8 | 8-bit f.p. [1], | 8-bit f.p. | IFX, TFLM |
| int16x8 | 16-bit f.p. | 8-bit f.p. | IFX |
| int16x16 | 16-bit f.p. | 16-bit f.p. | IFX |
| float | fl.p. [2] | fl.p. | IFX, TFLM |

[1] *f.p.*—fixed point. [2] *fl.p.*—floating point

## 5. Results and Discussion

To validate the experiment, the models were tested both on the desktop and on the target device to measure their computational burden and performance. Initially, the pretrained models had to be uploaded to the MTB-ML configurator 2.0 software, and their quantized versions were obtained. The achieved computational and memory requirements for the reduced models can be seen in Table 5 based on discharge profile 1 (the results are the same for profile 2). It is well noticeable that the 16-bit operations are more computationally efficient, with the int16x16 quantized model boasting the lowest number of cycles for both the ANN and CNN. Regarding the memory requirements, as expected, utilizing 8-bit information for data representation significantly reduces the amount of memory needed

both in terms of the solid flash memory (for model weight and bias matrices) and the temporary SRAM (scratch) for running inference. It is clearly visible that the ANN model uses around a quarter of the memory of the CNN, which is proportional to the number of trainable parameters of both models.

**Table 5.** Memory and computational burden analysis.

| Model | Quantization Type | Number of Cycles | Model Weights and Biases (kB) | Scratch Memory (kB) |
|---|---|---|---|---|
| CNN | int16x16 | 134,790 | 47.99 | 3.69 |
| | int16x8 | 201,932 | 24.04 | 3.69 |
| | int8x8 | 155,008 | 24.04 | 2.32 |
| ANN | int16x16 | 20,246 | 12.66 | 0.55 |
| | int16x8 | 30,878 | 6.36 | 0.55 |
| | int8x8 | 23,283 | 6.36 | 0.35 |

Considering the accuracy, in Table 6, the detailed results for both tested cycles can be seen divided between different quantization types and models. Consistent with previous findings [36], the models show no accuracy drop when used on the target device compared to the desktop validation prior to deployment. Therefore, only a single result is listed for each version. The models were validated both on the desktop and on the device, but only on-device results are presented. Generally, the reference model performance (not quantized) is significantly better considering the early life-cycle charging cycle 100, with errors not exceeding 3% in terms of the MAE. At that point of the lifecycle, the battery is still in the early cycles (linear phase) and has not degraded much; as a result, the behavior of the battery is more stable and can be accurately predicted based on the sensor measurements of V, I, and T. Cycle 400 is located close to the battery 'knee' point (as can be seen in Figure 7), where the behavior becomes unpredictable, which impedes accurate generalization for ML and DL models. In both cases, the ANN surpasses the CNN in terms of performance with a lower error in terms of both MAE and RMSE. For the quantized models, the best results are achieved by the ANN int16x16 with an MAE equal to 2.81% and an RMSE equal to 3.86%, which is around 0.5% better in comparison to the CNN at the same quantization. Moreover, considering cycle 100, which is the main use scenario, for the ANN, the differences between errors for different quantization bit precisions are significantly smaller than for the CNN. The difference does not surpass 0.2% in terms of the MAE, while for the CNN, the difference between int16x16 (best performing) and int8x8 (worst performing) exceeds 0.6%. The situation is inverted in the edge case (cycle 400) with an error difference of 1.5% for the ANN compared to 0.8% for the CNN.

**Table 6.** Performance analysis for the tested models in terms of MAE and RMSE—the best model is marked in bold.

| Cycle | Model | ANN | | CNN | |
|---|---|---|---|---|---|
| | | MAE | RMSE | MAE | RMSE |
| **Cycle 100** | Ref. model | 0.0269 | 0.0379 | 0.0295 | 0.0402 |
| | int8x8 | 0.0282 | 0.0389 | 0.0407 | 0.0493 |
| | int16x8 | 0.0303 | 0.0403 | 0.0379 | 0.0464 |
| | **int16x16** | **0.0281** | **0.0386** | **0.0342** | **0.0432** |
| **Cycle 400** | Ref. model | 0.0509 | 0.0662 | 0.0518 | 0.0666 |
| | int8x8 | 0.053 | 0.067 | 0.0499 | 0.0614 |
| | int16x8 | 0.0629 | 0.076 | 0.0481 | 0.0619 |
| | int16x16 | 0.0484 | 0.0638 | 0.0565 | 0.0707 |

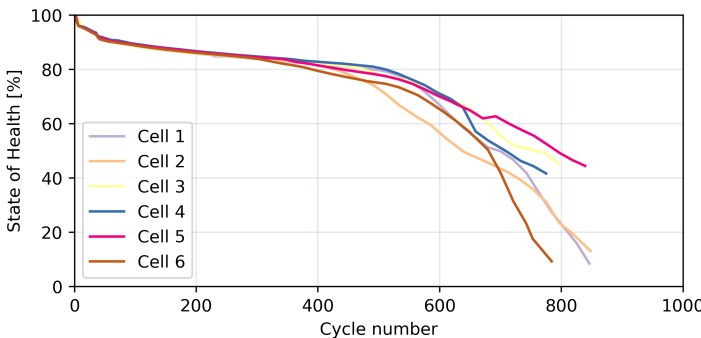

**Figure 7.** State of Health of the six studied cells. Around cycle 400, the state of the battery starts to degrade drastically.

For cycle 100, the quantized models behave as expected, with slight accuracy drops compared to the reference model at each further quantization step, which is particularly true for the CNN. However, in the case of cycle 400, a slight improvement in prediction accuracy over the reference model can be noted in some quantization options. This phenomenon can be further examined using graphs presented in Figures 8 and 9, which present the test error variation between the reference non-quantized models and their quantized counterparts, defined as:

$$Relative\ accuracy = (1 - \frac{Error_{quantized}}{Error_{reference}}) \cdot 100\% \tag{9}$$

In the case of cycle 100, the model follows the expected pattern with the lowest accuracy drop achieved for int16x16 quantization, which is equal to 4% for the ANN in terms of the MAE and 16% for the CNN. Due to the model complexity being connected to a significantly higher number of parameters, the accuracy drop is naturally higher for the CNN model as the information loss due to a lower bit precision is relevantly more significant. The same cannot be said about cycle 400, as it can be observed that the model behavior is inconsistent with the information loss due to conversion. This can be justified again by the unpredictable behavior of batteries in the later life stages, which implausibly might converge well with the decreased generalization precision learned from the training dataset. Hence, this situation should be seen as an exception that is caused by an artifact present in the test data and should not be seen as the expected pattern of behavior.

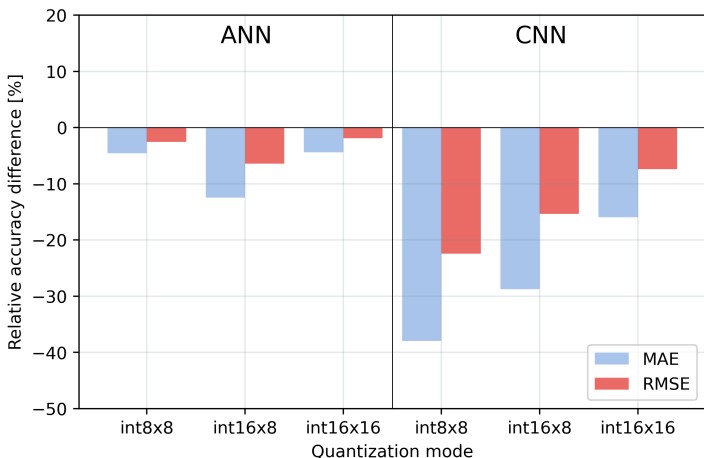

**Figure 8.** Cycle 100—comparison between the reference model and the quantized model performance.

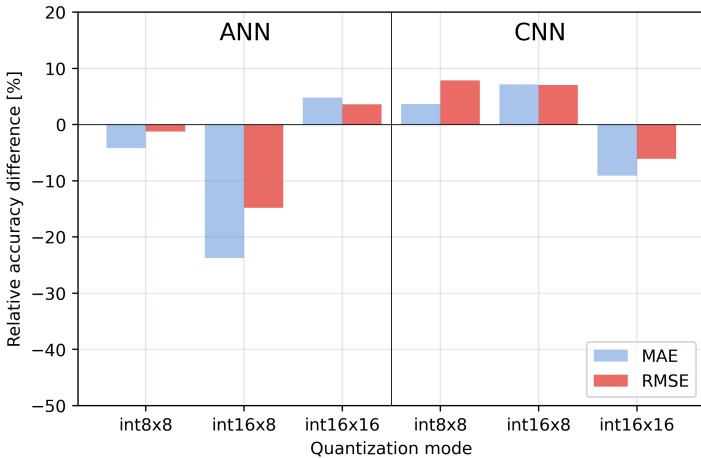

**Figure 9.** Cycle 400—comparison between the reference model and the quantized model performance.

The inconsistencies can be further studied by examining the quantization noise corresponding to each model and each quantization precision. In Figure 10, the model degradation due to quantization is studied by comparing the reference model prediction with quantized versions in terms of the MAE. It is easily visible that the int16x16 achieves a performance that is the most comparable to the original model trained and tested on a desktop with a quantization error not surpassing 0.5% in terms of the MAE. The error grows to 1.5% for the mixed quantization and to 0.6% for int8x8. This behavior is consistent with a higher bit precision and lower reductions in the model weights. Moreover, it is highly desirable, as with PTQ, most of the model training, validation, and hyperparameter tweaking is performed before quantization; therefore, any deviation may invalidate previous fine-tuning [46].

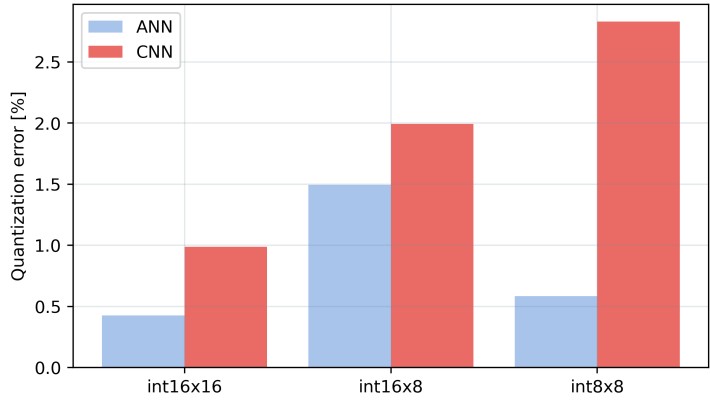

**Figure 10.** Mean quantization error for each tested model. The mean was calculated based on four test runs of each model for different test cycles.

Since the initial model design prioritized TinyML implementation, the pre-quantized model sizes are inherently small. This characteristic allows for less aggressive quantization, focusing on balancing memory efficiency with minimal performance degradation. In this case, int16x16 quantization reduces the model weight size by over 50% compared to the original precision (e.g., float32) for both the CNN and ANN models. This translates to a memory footprint of less than 10% for the CNN and less than 5% for the ANN on the flash memory of the target device. Furthermore, in Figure 10, it can be seen that int16x16 quantization typically introduces a minimal accuracy loss, with errors not exceeding 0.5% compared to the pre-quantized model. This ensures that the model maintains its effectiveness in real-world applications.

Considering both memory efficiency and performance, the int16x16 ANN model emerges as the optimal choice for deployment and further study. Its compact size allows for efficient operation on resource-constrained IoT devices, while the minimal performance degradation ensures accurate battery SoC estimation. This selection prioritizes the key considerations for TinyML applications on resource-limited devices.

Regardless of the results, there is still an issue with a declining performance in the later stages of battery life. Despite the satisfactory results obtained by the tested models, a good prediction accuracy should be independent of the aging phase of the battery and the test device. Therefore, in future works, it is necessary to test other common architectures for time-series analysis, including models like CNN-LSTM or CNN-GRU. A robust selection procedure should be performed considering a trade-off between the accuracy and computational and memory requirements of the model.

## 6. Conclusions

TinyML presents a new and exciting frontier for the development of smart measuring devices that are capable of aggregating data and making decisions on the edge. This paper contributes to the state of the art by presenting a robust case study on IoT implementation of DL algorithms for SoC estimation. A new, very large open-access battery dataset published by TUB-PoliMi has been utilized to train the proposed models and test the developed methodology. The dataset is based on LG 2.5Ah 18650 batteries of NMC type and contains information on the voltage, current, and temperature of the cells. Various driving profiles are tested at different discharge temperatures ranging from −20 °C to 35 °C. These new data allow for a detailed comparison of battery behavior, as the models could be tested using a different battery cell and various driving cycles.

Provided with the dataset, two different DL models are trained, quantized through PTQ, and subsequently deployed on the CY8CPROTO-062S3-4343W PSoC 62S3 target device. The selected quantization infrastructure supports different quantization precision values. This allows for demonstrating the benefits of using a less destructive precision—int16x16 compared to the standard int8x8 supported by the majority of open-source frameworks such as TFLM or ONNX. With int16x16, the models suffer from less quantization noise, making them more comparable to the reference models trained and tested on the desktop before PTQ. Certainly, the price to pay is the increased on-device memory footprint of the deployed model. However, with models such as the demonstrated ANN, the memory occupied by the model weights is less than 3% of the total available flash memory. In the end, the best result in terms of the MAE was achieved by the ANN in the early cycles with a 2.81% error for an already quantized model.

In the tests, the ANN outperforms the more complex CNN in terms of both inference time and accuracy, which is in line with the previous findings in [12,35,36], where often simpler models achieved comparable or better results than their more advanced counterparts. Moreover, the error achieved by the proposed ANN approach is lower compared to those found in related research papers. Despite the satisfactory results of the presented models, it would be necessary to evaluate other popular DL architectures that are regularly used for SoC prediction, such as LSTM, GRU, CNN-LSTM, and CNN-GRU, in future research. These advanced DL models for time-series analysis could deeply benefit from being paired with a more robust and detailed dataset from TUB-PoliMi [39]. Moreover, in future works, research can be extended to add additional steps and secondary prediction models that could focus on tasks such as SoH estimation or RUL trajectory prediction, which are vital for prognostics, diagnostics, and general health management of batteries. Moreover, some further experimentation will be performed to enrich the training dataset, such as the implementation of time-series data augmentation techniques to address the inconsistencies in the late cycle.

**Author Contributions:** Conceptualization, E.O. and S.L.; methodology, S.G.; software, S.G. and M.S.; validation, S.B. and F.R.; formal analysis, S.G.; investigation, S.G. and M.S.; resources, S.B. and F.R.; data curation, S.G.; writing—original draft preparation, S.G. and M.S.; writing—review and editing,

S.G., M.S. and E.O.; visualization, M.S.; supervision, S.L.; project administration, E.O.; funding acquisition, E.O. All authors have read and agreed to the published version of the manuscript.

**Funding:** This study was carried out within the MOST—Sustainable Mobility Center and received funding from the European Union Next-GenerationEU (PIANO NAZIONALE DI RIPRESA E RE-SILIENZA (PNRR)—MISSIONE 4 COMPONENTE 2, INVESTIMENTO 1.4—D.D. 1033 17/06/2022, CN00000023). This manuscript reflects only the authors' views and opinions; neither the European Union nor the European Commission can be considered responsible for them.

**Data Availability Statement:** The data presented in this study are available in this article.

**Conflicts of Interest:** Authors Susheel Badha and Filippo Rosetti were employed by the company Infineon Technologies. The remaining authors declare that the research was conducted in the absence of any commercial or financial relationships that could be construed as a potential conflict of interest.

## Abbreviations

The following abbreviations are used in this manuscript:

| | |
|---|---|
| ANN | Artificial Neural Network |
| BMS | Battery Management System |
| DL | Deep Learning |
| GRU | Gated Recurrent Unit cell |
| IoT | Internet of Things |
| Li-Ion | Lithium-Ion |
| LSTM | Long Short-Term Memory cell |
| MAE | Mean Absolute Error |
| ML | Machine Learning |
| MTB-ML | ModusToolbox Machine Learning Configurator |
| PTQ | Post-Training Quantization |
| QAT | Quantization-Aware Training |
| RMSE | Root Mean Square Error |
| RUL | Remaining Useful Life |
| SoC | State of Charge |
| SoH | State of Health |
| TinyML | Tiny Machine Learning |

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
