# Peer review of "A Case Study of a Tiny Machine Learning Application for Battery State-of-Charge Estimation"

_electronics, doi:10.3390/electronics13101964_

Round 1
Reviewer 1 Report
Comments and Suggestions for Authors
The paper considers TinyML techniques to predict the SoC of embedded devices, which is interesting. However, there is a lack of details differentiating with related work and some results need to be extended for improved contribution. In addition, please consider making the dataset open source, as these results are not reproducible without it (open-sourcing the model is one important part, the other is the used data), making the contribution of the paper very limited.
I have the following remaining remarks and suggested additions:
- Q is not defined on page 1
- Please discuss or verify (with reference or own results) whether ML is less complex than "complex mathematical relations and electrochemical principles" in model-based approaches
- How does the work presented in this paper compare to the related work presented in [7, 21, 22]. Please add a quantitative comparison (maybe in the form of a table) with metrics such as, accuracy, complexity, model used, datasets used, etc.
- Can the authors please provide a reason why a new private dataset was necessary. Please consider making datasets open source when presenting data-driven and ML-driven results, in order to verify and reproduce your results.
- In table 1, a sampling rate of "0.1 seconds" is mentioned, I think the authors meant 10 Hz (as seconds is not a rate).
- Can the authors also mention the input size of the model. I suppose the input is first flattened from (600x3), after which "(34,8,1)" neurons are added?
- Why are the on desktop and target accuracy results both presented? No difference in accuracy is expected between the two of them. The authors can mention this in text and remove redundant information in table 5.
- Considering the unexpected increase in accuracy for CNN on integer quantized models: this could be a random bias, please consider adding results for other unseen battery cells and / or add cross-validation across these cells.
- I'm missing a discussion / results on the ideal input size (which is now 1 minute). This could reveal whether the accuracy can be even more increased with longer input samples (at the expense of a higher prediction latency)
- Is quantization aware training considered? I assume only post-training quantization is performed in this paper; please state this.
- Table 3 mentions multiple inference engines, but it is not clear which ones are used for the results. Is there any difference; does IFX apply other optimizations?
Comments on the Quality of English Language- "Tab. ??" reference is missing on page 7
Reviewer 2 Report
Comments and Suggestions for Authors
The main objective of this paper is research on the application of the DL algorithm to estimate the SoC of batteries in IoT devices. Two NN architectures, namely CNN and ANN, were trained and tested. For this purpose, the authors create their own database of the main parameters of batteries.
The created models are trained offline (on desktop), then quantized and optimized, with the main goal of being tested on a specific IoT device.
The research enables a better assessment of the potential of using IoT devices for SoC predictions.
The original part of the research Is related to:
1. The authors have created their own battery parameter dataset recorded at different temperatures.
2. Based on this dataset DL algorithms for SoS estimation have been implemented in IoT devises.
3. The achieved results considering the accuracy of the trained models are more than satisfactory and can be used as a solid ground for conducting further studies on the topic.
The following improvements can be made to make the research more clear and completed:
1. It would be necessary to evaluate other popular DL architectures that are regularly used for SoC prediction, such as LSTM, GRU, CNN-LSTM, and CNN-GRU.
2. In future works the research can be extended to add additional steps and secondary prediction models that could focus on tasks such as SoH estimation or RUL trajectory prediction.
3. Even more complex datasets can be used, using different discharging cycles and battery types to improve the model’s performance and robustness.
The conclusion is made correctly. The main questions are precisely defined and proper experiments are fulfilled.
The references are appropriate to the research.
Reviewer 3 Report
Comments and Suggestions for Authors
The authors present a case study on IoT implementation of DL algorithms for SoC estimation. The achieved results including the accuracy of the trained models, the memory use, and the computational load can be used for conducting further studies on this topic. The research can be extended to add additional steps and secondary prediction models that could focus on tasks such as SoH estimation or RUL trajectory prediction. This work can help the development of battery management systems in the industry for the battery pack of electric vehicles and large-scale energy storage systems. Therefore, I think this work is appropriate to be published on electronics. Before that, I have some questions and suggestions:
1. Line 29: Please polish this sentence.
2. Line 60: The word, "itself", can be deleted.
3. Line 72: This format seems to be not scientific style. Please change that to “In a previous study [7]”.
4. Line 103: It is better to replace “constant-I constant-V” with "constant current and constant voltage procedures".
5. Figure 1: Please use “the 60th second”.
6. Line 103: Please replace “neurons’ output” with “output of neuron”. Please replace “is” with “are”.
7. Line 191: Please replace “100-th and 400-th cycles” with “the 100th and the 400 cycles”.
8. Figure 4: I think the x-axis is the time and the unit is the second (s). If it is, the "cycle sample" should be changed to “Discharge time”.
9. Line 206: Please check “Table 4”.
10. Figure 4 X-axis: Please delete "[-]". There is no unit for the cycle number.
11. Figure 4 Caption: Please replace “Around cycle 400 the state of the battery starts to
degrade drastically” with “At the 400th cycle, the state of the battery starts to degrade drastically”
12. All the Figure and table captions should have a period at the end of the sentences.
13. Please replace “between the model’s accuracy, and computational and memory requirements” with “between the accuracy, computational and memory requirements of the model”.
Reviewer 4 Report
Comments and Suggestions for Authors
While the topic of your research is highly relevant and of significant interest, the manuscript as it stands does not fully meet the required standards for publication due to several key issues. I recommend a comprehensive revision, taking into account the detailed feedback provided.
1. The need for accurate state of charge (SoC) prediction is mentioned as critical for energy conservation and battery life extension. However, the justification provided seems inadequate. Including specific examples or case studies where accurate SoC prediction has had a significant impact could strengthen your argument.
2. The introductory section lacks a comprehensive review of related work. Expanding on both model-based and data-driven methods with appropriate references to existing research would provide a solid foundation for your study. In addition, while the title of the paper suggests a case study approach to Tiny Machine Learning, it lacks a comprehensive analysis of different cases, which is essential for a full understanding of the topic.
3. The use of a private dataset raises concerns about the reproducibility and fairness of your research. For future studies, consider using publicly available datasets or providing a more detailed justification and ethical considerations for the use of proprietary data.
4. The case study section needs a more detailed description of the dataset and an explanation of the cases selected for analysis. Clearly stating why particular cases were selected and their relevance to the research objectives would clarify the intentions of your study for your readers.
5. The implementation of the Bayesian optimization approach and the development of a 1D CNN model are not sufficiently explained. Clarification of how these methods were applied and their underlying rationale would increase the technical depth of the manuscript.
6. The practicality of your model, particularly concerning the max-pooling operation in CNNs, is questioned. A deeper discussion of the choice of this technique and its impact on the results would be beneficial.
7. The manuscript discusses quantization mainly in terms of reducing computational complexity. However, the practical significance of this, especially in the context of SoC prediction where precision is paramount, seems to be underexplored. Expanding on how quantization affects the accuracy and reliability of SoC predictions would add value.
8. The interpretation of the results from the figures and tables presented is unclear. A more detailed analysis linking these results to your conclusions would improve understanding.
9. Inconsistencies in the performance analysis raise questions about the reliability of the method, especially in the later stages of battery life. Addressing these inconsistencies with more thorough analysis or further testing could strengthen the credibility of your findings.
10. The manuscript focuses largely on future research directions and limitations without sufficiently summarizing the key findings and their implications. A more conclusive summary of the results of your study and their significance in the field would provide a stronger conclusion.
Round 2
Reviewer 1 Report
Comments and Suggestions for Authors
The authors have addressed all my concerns. The paper is ready for publication in my opinion.
Author Response
Dear reviewer, thank you for accepting our paper for publication in MDPI Electronics. Your comments and suggestions proved invaluable in refining our work and crafting the best possible version of the manuscript.
In the updated version you can find some further refinements and clarifications regarding the dataset, the utilized models, utilized quantization, and results. We hope these changes improve the readability and clarity of the paper even further.
Best regards,
The authors.
Reviewer 4 Report
Comments and Suggestions for Authors
Thanks to the authors' diligent efforts, most of the issues I raised have been addressed. However, despite these revisions, some areas of the resubmitted manuscript still need further refinement. If the following improvements are made, I believe this paper would be suitable for publication in the Electronics journal.
1. While the importance of battery management systems (BMSs) is mentioned in the introduction, there is no clear explanation of how BMSs relate to the expansion of the electric vehicle industry. This would add to the overall context of the paper. In addition, there is a lack of or unclear explanation of battery metrics such as SoC, SoH, and RUL, which should be clarified to explain the difference between these metrics and why they need to be accurately estimated.
2. A balanced discussion of both model-based and data-driven approaches is needed in the literature review section. Highlighting the benefits and challenges of ML/DL models on edge devices, including the role of TinyML, will make the literature review section more complete.
3. Despite the authors' efforts, the dataset description is not comprehensive enough. The authors should clarify the relevance of battery type, conditions, and driving profile and add a description of the process of resampling to 1 Hz to address the size, complexity, and impact on model training and testing.
4. The description of the structure of the ANN and CNN in the model architecture is inadequate; in particular, providing background on PTQ and explaining how quantization affects memory capacity and model accuracy would improve understanding of the architecture description section.
5. The comparison of ANN and CNN models in the Results and Discussion section lacks mention of various aspects. You should add a comparison of various aspects, including accuracy, memory usage, computational load, etc. Also missing is an explanation of how the model handles SoC predictions at different stages of battery life. Make sure that the graphical material and textual explanations are consistent to help the reader understand.
6. The conclusion section lacks a summary of the dataset usage, model performance, and how TinyML contributes to SoC prediction. This would make the conclusions clearer. In addition, future work should explore other models (CNN-LSTM, CNN-GRU, etc.) and suggest ways to enrich the training dataset to reduce inaccuracies in the late cycle.
Comments on the Quality of English LanguageThere are some grammatical errors and inconsistencies in technical terminology that need to be corrected. In addition, strengthening the flow between sections, such as the transition from dataset description to model training, would improve the readability of the paper.
